# Development of sustainable alkali activated composite incorporated with sugarcane bagasse ash and polyvinyl alcohol fibers

**Munir Iqbal**[1]*, **Muhammad Ashraf**[1], **Loai Alkhattabi**[2], **Sohaib Nazar**[3¤], **Jihad Alam**[1], **Hisham Alabduljabbar**[4], **Ali Husnain**[1]

1 Department of Civil Engineering, Ghulam Ishaq Khan Institute of Engineering Sciences and Technology, Topi, Pakistan, 2 University of Jeddah, College of Engineering, Department of Civil and Environmental Engineering, Jeddah, Saudi Arabia, 3 Shanghai Key Laboratory for Digital Maintenance of Buildings and Infrastructure, School of Naval Architecture, Ocean and Civil Engineering, Shanghai Jiao Tong University, Shanghai, China, 4 Department of Civil Engineering, College of Engineering in Al-Kharj, Prince Sattam Bin Abdulaziz University, Al-Kharj, Saudi Arabia

¤ Current address: Department of Civil Engineering, Comsats University Islamabad-Abbottabad Campus, Islamabad, Pakistan

* muniriqbal0345@gmail.com

## Abstract

The infrastructure boom has driven up cement demand to 30 billion tons annually. To address this and promote sustainable construction, researchers are developing solutions for carbon-neutral building practices, aiming to transform industrial waste into an eco-friendly alternative. This study aims to develop and enhance the mechanical and durability properties of alkali-activated composites (AACs) by incorporating varying amounts (5, 10, 15, and 20%) of finely ground bagasse ash (GBA) and polyvinyl alcohol (PVA) fibers. Results indicate that higher GBA content initially reduces the 7th and 14th-day strength but results in increased strength at later ages. The optimum 28-day strength is achieved with a 10% GBA content, leading to a 10% increase in compressive strength, 8% increase in tensile strength, and 12% increase in flexural strength. Additionally, the incorporation of GBA enhanced the resistance of the composite to chloride ingress, thus reducing its conductance and increasing the overall durability. This study demonstrated the potential of GBA as an eco-friendly material, emphasizing the significance of tailored AACs formulations for durable and sustainable construction practices.

## Introduction

Ordinary Portland cement (OPC)-based concrete is a durable, affordable, and widely available material that exhibits a high compressive strength, making it a preferred choice for use in civil infrastructure. However, cement factories produce enormous amounts of carbon dioxide ($CO_2$) gas and are responsible for 5–8% of global $CO_2$ emissions [1, 2]. To achieve a safe and sustainable approach, it is imperative to minimize the $CO_2$ emissions and energy expenditures associated with cement production. For this purpose, an alkali-activated composites (AACs)

**Data Availability Statement:** All relevant data are within the manuscript and its Supporting Information files.

**Funding:** The author(s) received no specific funding for this work.

**Competing interests:** The authors have declared that no competing interests exist.

with low carbon footprint were developed [3]. AACs provide favorable attributes, including their lightweight nature, reduced carbon emissions, inherent self-healing properties, commendable impact resistance, and enhanced fire resistance [4, 5]. Further, the incorporation of dispersed fibers in the AAC matrix has been observed to improve the tensile properties and hinder the propagation of microcracks to macrocracking, effectively preventing brittle failure [3, 6]. Different types of fibers, including steel, glass, basalt, and polyvinyl alcohol (PVA), have found application in fiber-reinforced concrete (FRC) [7–11]. Each fiber type possesses distinct properties that significantly influence various aspects of performance such as tensile strength, ductility index, and post-cracking behavior [12, 13]. Basalt fibers have been noted to enhance compressive strength [14], whereas steel fibers improve tensile strength and offer good resistance to freeze-thaw cycles. However, steel fibers pose challenges related to cost, workability, and durability [15–17]. Glass fibers, on the other hand, suffer from instability in alkaline environments [18, 19]. PVA fibers emerge as particularly suitable for enhancing overall concrete performance due to their stability in alkaline environments, high tensile strength, and higher elastic modulus values [20]. Consequently, they contribute to improved mechanical characteristics and overall concrete performance [21, 22].

There is an increasing trend in the use of industrial waste materials like fly ash, silica fume, and ground granulated blast furnace slag (GGBS) for manufacturing AACs [23–27]. Wheat straw ash, palm oil fuel ash, rice husk ash, and sugarcane bagasse ash (SBA) are some of the more examples of biowastes that find the same use [28–32]. The inherent amorphous nature of these minerals and the higher amount of silicon dioxide ($SiO_2$) play a crucial role in their stability and contribute to the development of high-strength AACs [33–35].

Among the various types of waste discussed earlier, sugarcane bagasse stands out as particularly significant [28, 36–38]. It is the fibrous residue remaining after extracting juice from sugarcane and accounts for a substantial portion of global sugarcane cultivation, around 40–50% (See Fig 1) [34, 39] Upon combustion, it generates a high amount of ash, which finds applications in materials such as sodium water glass and ceramics [40, 41]. Due to its pozzolanic activity and abundant silica-alumina content, SBA holds promise in the construction industry [28, 42, 43]. Studies have shown that incorporating ground bagasse ash (GBA), a grinded form of SBA, into concrete enhances its physical, mechanical, and durability properties [40, 43–48]. For example, research by Shafiq et al. [49] demonstrated a significant increase in concrete strength with the addition of 20% GBA content. Even smaller proportions, around 5%, have shown benefits, as reported in other studies [50]. However, research by Kawada et al. [51] suggests that incorporating up to 15% GBA has minimal impact on core concrete properties. This finding is supported by Ganesan et al. [11], who replaced 20% of cement with GBA. Additionally, preliminary studies have demonstrated the feasibility of using GBA as a precursor for alkali-activated binders [52–56]. For instance, Castaldelli et al. [57] and Pereira et al. [58] examined the partial replacement of GGBS by SBA, while Tippayasam et al. [59] and Castaldelli et al. [60] explored binary systems of fly ash and SBA. Yadav et al. [61] investigated the partial replacement of metakaolin (MK) by SBA, and Tchakouté et al. [62] assessed the use of sodium waterglass from SBA as an activator for producing MK-based geopolymers. Recently, Akbar [63] utilized SBA exclusively as a precursor for geopolymer production reinforced with propylene fibers. Despite these advancements, the application of GBA, particularly untreated GBA, in more complex alkali-activated composite (AAC) systems remains underexplored.

While GBA in concrete and geopolymer formulations has been extensively researched (see Table 1), its efficiency in AACs containing fly ash, GGBS, and polyvinyl alcohol (PVA) fibers is still a relatively new topic that warrants further exploration. The current study addresses this gap by developing and evaluating the performance of AACs incorporating GGBS, fly ash, PVA fibers, and varying proportions of untreated GBA. The primary objective is to determine the

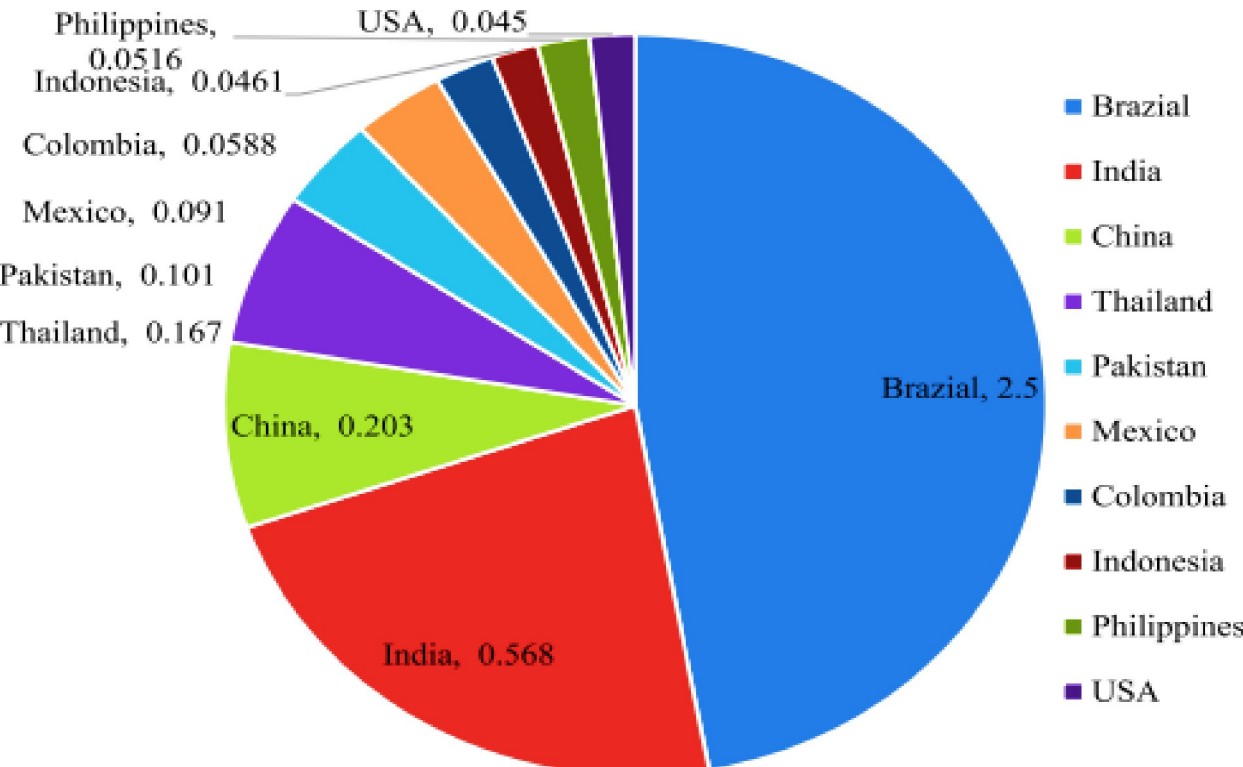

**Fig 1. SBA production in different countries (million tons).** Reprinted from [64] under a CC BY license, with permission from construction and building materials, original copyright [2020].

optimal GBA content that maximizes improvements in mechanical performance and durability of AACs, while also considering economic viability and environmental sustainability. This research aims to provide a comprehensive understanding of the potential benefits and limitations of GBA in advanced AAC formulations, thereby contributing to the development of more sustainable and high-performance construction materials.

## Materials

### Class-F fly ash, bagasse ash and GGBS

Class-F fly ash, made from anthracite, was used in this study. This fly ash had a calcium content of less than 5%. The particle size distribution curve of fly ash is shown in Fig 2.

The collected bagasse ash was burned at 700°C for 90 min and allowed to cool to room temperature in open air. Once cooled, the ash was stored in airtight containers for further analysis. To reduce the presence of carbon-rich amorphous fibrous particles that contain little crystalline silica, sieving was done with a 300-μm sieve. The sieving process allowed the separation of fine particles which are to be used for subsequent testing and utilization. According to another study [42], after sieving, burned bagasse ash is left with fine particles that are rich in silica.

This study utilized GGBS as 20% of the total binder. The specific gravity of the GGBS is found to be 2.90, while its water absorption is 1.38%. The results of the X-ray fluorescence (XRF) test performed to determine the composition of the binder elements are shown in Table 2.

**Table 1. Hardened properties of concrete at various substitution levels of GBA.**

| GBA | Mix | W/B ratio | SP % | Compressive strength (MPa) 28 days | 28-day splitting tensile strength (MPa) | 28-day flexural strength (MPa) | Ref. | GBA (%) | Mix | W/B ratio | SP % | Compressive strength (MPa) 28 days | 28-day splitting tensile strength (MPa) | Ref. |
|---|---|---|---|---|---|---|---|---|---|---|---|---|---|---|
| 0 | M25 | 0.44–0.63 | – | 24.45 | 2.8 | 4.1 | [65] | 0 | M30 | Not Given | – | 37.2 | 3.2 | [66] |
| 10 | | | | 24.45 | 3.1 | 5.4 | | 5 | | | | 38.7 | 3.3 | |
| 15 | | | | 24.9 | 3 | 6.3 | | 10 | | | | 40.3 | 3.4 | |
| 20 | | | | 19 | 1.55 | 4.2 | | 15 | | | | 22 | 2.7 | |
| 25 | | | | 18.6 | 1.4 | 3.6 | | 20 | | | | 19.6 | 2.5 | |
| 0 | M35 | 0.45–0.55 | – | 18.3 | 1.1 | 2.5 | [64] | 0 | M20 1:1.75:2.89 | 0.425–0.5 | – | 26.9 | 1.6 | [67] |
| 10 | | | | 31.1 | 3.25 | 5.1 | | 5 | | | | 26.5 | 2 | |
| 15 | | | | 32 | 3.25 | 6 | | 10 | | | | 27.8 | 2.5 | |
| 20 | | | | 32 | 3.4 | 6.5 | | 15 | | | | 29.4 | 2 | |
| 25 | | | | 30 | 2.7 | 4.45 | | 20 | | | | 26.8 | 1.9 | |
| 30 | | | | 29 | 2.1 | 3.9 | | 0 | | | | 28.9 | 2 | |
| 0 | M20 | 0.48 | – | 21.5 | 1.5 | 3.5 | [64] | 10 | M25 | Not Given | 0.5 | 28.8 | 2.7 | [67] |
| 5 | | | | 29.5 | 1.9 | 3.7 | | 15 | | | | 29.9 | 2.1 | |
| 10 | | | | 24.7 | 1.6 | 3.6 | | 20 | | | | 27.9 | 2.2 | |
| 15 | | | | 19.3 | 1.5 | 3.4 | | 0 | | | | 29 | 4 | |
| 20 | | | | 18.9 | 1.3 | 3.2 | | 3 | | | | 33 | 4.9 | |
| 25 | | | | 17.7 | 1.2 | 3 | | 6 | | | | 34.4 | 5.5 | |
| 0 | M30 1:1.52:2.82 | 0.42 | – | 39.5 | 1.3 | 3.9 | [64] | 9 | | | – | 34 | 5 | [64] |
| 5 | | | | 45.5 | 1.4 | 4.4 | | 12 | | | | 27 | 3.8 | |
| 0 | M25 1:1.62:3.02 | 0.5 | – | 43 | 1.3 | 4.1 | [51] | 0 | M25 1:2.35:3.01 | 0.44–0.5 | – | 34.1 | 3.4 | [68] |
| 5 | | | | 39.7 | 1.3 | 4 | | 5 | | | | 32.5 | 3.3 | |
| 10 | | | | 30.1 | 1.1 | 2.9 | | 10 | | | | 30.1 | 3.2 | |
| 15 | | | | 18.7 | 0.9 | 2 | | 15 | | | | 28.5 | 3 | |
| 20 | | | | 33.3 | 2.1 | 6.1 | | 20 | | | | 25.6 | 2.9 | |
| 25 | | | | 37.4 | 2.8 | 6.7 | | 25 | | | | 36.9 | – | |
| 0 | 1:1.59:2.63 | 0.4 | – | 39.5 | 2.6 | 4.6 | [69] | 0 | M25 | 0.45–0.46 | – | 33.8 | 3.4 | [68] |
| 5 | | | | 40 | 2.9 | 4.8 | | 5 | | | | 32.1 | 3.3 | |
| 10 | | | | 38 | 2.4 | 4.3 | | 10 | | | | 30.5 | 3.2 | |
| 15 | | | | 37.6 | 2.1 | 4.2 | | 15 | | | | 28.1 | 3 | |
| 20 | | | | 37 | 1.8 | 4.1 | | 20 | | | | 25.4 | 2.8 | |
| 25 | | | | 36 | 1.7 | 3.9 | | 25 | | | | 32.4 | 3.4 | |
| 0 | M25 1:2.11:3.275 | 0.42 | | 36.18 | 2.55 | 5.9 | [70] | 5 | M35 1:1.164:2.733 | 0.41 | – | 30.9 | 3.3 | [71] |
| 5 | | | | 36.89 | 2.59 | 6.1 | | 10 | | | | 29.3 | 3.2 | |
| 10 | | | | 37.52 | 2.75 | 6.45 | | 15 | | | | 28 | 3 | |
| 15 | | | | 33.93 | 2.25 | 5.8 | | 20 | | | | 24.3 | 2.8 | |
| 20 | | | | 30.07 | 1.92 | 4.9 | | 0 | | | | 45.62 | - | |
| 25 | | | | 24.85 | 1.76 | 4.1 | | 25 | | | | 46.75 | – | |
| 0 | 1:1.49:1.89 | 0.38 | 0.6 | 62 | – | – | [72] | 0 | M20 1:1.9998:3.37 | Not Given | – | 47.81 | – | [64] |
| 5 | | | | 91 | – | – | | 5 | | | | 45.53 | | |
| 10 | | | | 88 | – | – | | 10 | | | | 44.46 | | |
| 15 | | | | 84 | – | – | | 15 | | | | 20 | | |
| 20 | | | | 65 | – | – | | 20 | | | | 21.33 | | |
| 25 | | | | 61 | – | – | | 25 | | | | 20.67 | | |
| 30 | | | | 53 | – | – | | 30 | | | | 20.89 | | |
| | | | | 37 | – | – | | 35 | | | | 22.67 | | |

(*Continued*)

**Table 1.** (Continued)

| GBA | Mix | W/B ratio | SP % | Compressive strength (MPa) 28 days | 28-day splitting tensile strength (MPa) | 28-day flexural strength (MPa) | Ref. | GBA (%) | Mix | W/B ratio | SP % | Compressive strength (MPa) 28 days | 28-day splitting tensile strength (MPa) | Ref. |
|---|---|---|---|---|---|---|---|---|---|---|---|---|---|---|
| 0 | M25 1:1.5:3 | 0.53 | – | 41 | 4.5 | – | [28] | 0 | M20 1:2.25:2.86 | 0.55 | – | 27.93 | – | [73] |
| 5 | | | | 42 | 4.75 | – | | 5 | | | | 28.34 | | |
| 10 | | | | 41 | 4.85 | – | | 10 | | | | 24.45 | | |
| 15 | | | | 39 | 4.95 | – | | 15 | | | | 22.01 | | |
| 20 | | | | 33 | 4.8 | – | | 20 | | | | 19.71 | | |
| 25 | | | | 30 | 4 | – | | 25 | | | | 24 | | |

## PVA fibers

PVA fibers with a specific gravity of 1.3 are used in this study. These fibers possess a high tensile strength and exhibit good resistance to chemicals and alkalis, making them suitable for alkali-activated applications. They effectively bond with the alkali-activated matrix, enhancing the load transfer and preventing fiber pull-out. Table 3 lists the properties of the PVA fibers obtained from the manufacturer.

## Fine aggregate

Surface saturated river sand was used as the fine aggregate. Sieving was performed according to the ASTM standard [74]. The surface moisture content was 2.81%. The specific gravity and water absorption following the ASTM [75] procedure were 2.6 and 3%, respectively.

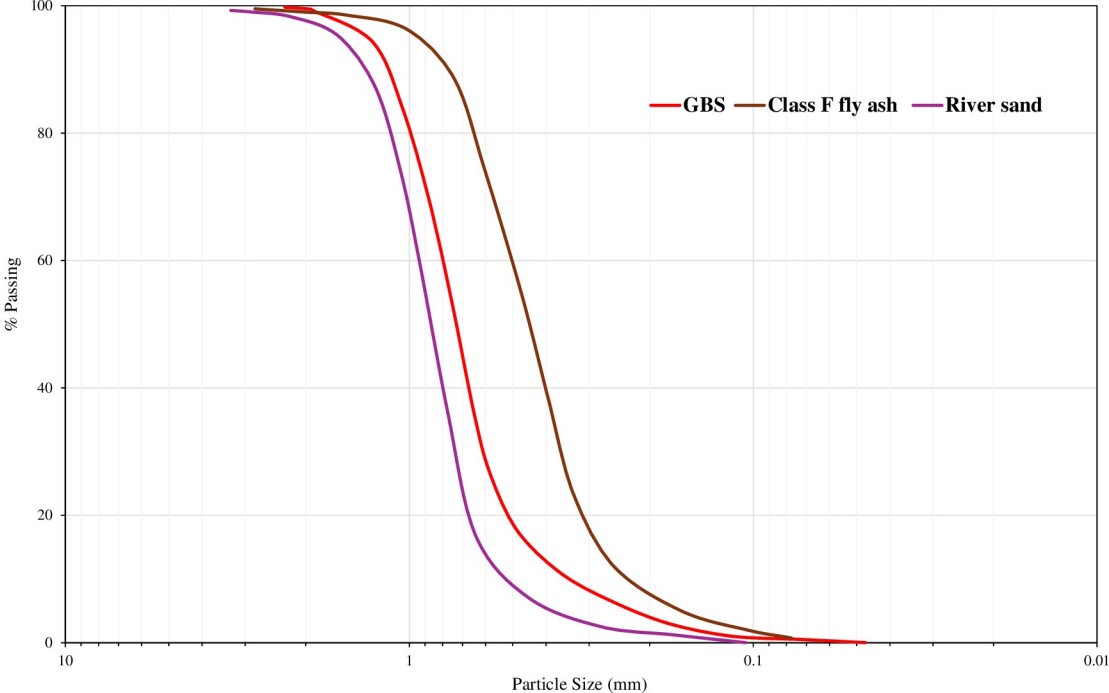

**Fig 2. Particle size distribution curve.**

**Table 2. Chemical composition of cement, GBA, and fly ash.**

| Oxide Composition | SiO$_2$ | Fe$_2$O$_3$ | Al$_2$O$_3$ | CaO | Na$_2$O$_3$ | MgO | SO$_3$ | LOI | Blaine fineness cm$^2$/g | Density g/cm$^3$ | S. G |
|---|---|---|---|---|---|---|---|---|---|---|---|
| GGBS | 42.3 | 3 | 12 | 40 | 0.95 | 1.4 | 2 | 2.2 | 4000 | 2.8 | 2.9 |
| GBA | 66.3 | 1.5 | 9.3 | 10.1 | 4.2 | — | 7.2 | 7.1 | 2849 | 2.3 | 2.2 |
| Fly ash | 51.3 | 25.25 | 12.8 | 0.81 | 0.74 | 1.5 | 0.2 | 0.54 | 2920 | 2.6 | 2.5 |

## Activators and superplasticizer

In this study, the alkaline activator was prepared by mixing sodium silicate (Na$_2$SiO$_3$) solution and sodium hydroxide (NaOH) solution. The Na$_2$SiO$_3$ solution comprised of 14.7% sodium oxide (Na$_2$O), 29.4% silicate (SiO$_2$), and 44.1% solids. The NaOH solution with a molarity of 14 was obtained by dissolving solid granulated caustic soda in tap water.

To induce flowability and reduce slump retention, a sulfonated naphthalene-based water-reducing agent conforming to ASTM C—494 Type G was used as a superplasticizer (SP). It was a liquid with a dark brown color and had a density of 1.13 ± 0.03 g/cm$^3$. The recommended dosage ranged from 0.3% to 2%. This SP contained no chlorides, ensuring it was suitable for use in reinforced concrete without the risk of corrosion. It had a pH value of 4.8 and a solids content of 30%.

## Procedure and methodology

The outline of the current investigation is depicted in Fig 3.

### Grinding of the GBA

The grinding process involved mechanical treatment in a ball mill. The objective of grinding was to increase the surface area of the GBA. Throughout the one hour grinding process, a constant grinding speed of 66 revolutions per minute was maintained, while the ratio of grinding media to ash was maintained at 5:1 by weight (i.e., 5 kg of balls to 1 kg of ash). The above procedure is adopted from another studies [76].

### Determination of fineness and pozzolanic reactivity of GBA

The fineness of the GBA was determined using a Blaine air permeability apparatus in accordance with ASTM C204–11 [77]. The desired surface area was obtained after 60 min of continuous grinding. Subsequently, the pozzolanic reactivity of GBA was evaluated using the Chappelle test, following the guidelines outlined in NF P 18–513 [78–80]. A Chappelle test quantifies the reduction in Ca (OH)$_2$ caused by its interaction with siliceous or aluminosilicate compounds within the ash. The results met the minimum requirements (330 mg of CaO/g of pozzolan). This indicates that GBA possesses favorable chemical reactivity. The outline of the extraction of SBA from sugarcane is shown in Fig 4.

### Scanning electron microscopy (SEM) of the bagasse ash

A microstructural investigation of several samples of GBA was undertaken by Nasir et al. Fig 5 displays the images of the examined samples at two magnification levels: X$_{500}$ and X$_{3000}$.

**Table 3. Properties of PVA fiber.**

| Fiber type | Length of fiber (mm) | Diameter (mm) | Tensile strength (MPa) | Elastic modulus (GPa) |
|---|---|---|---|---|
| PVA | 12 | 0.04 | 1600 | 40 |

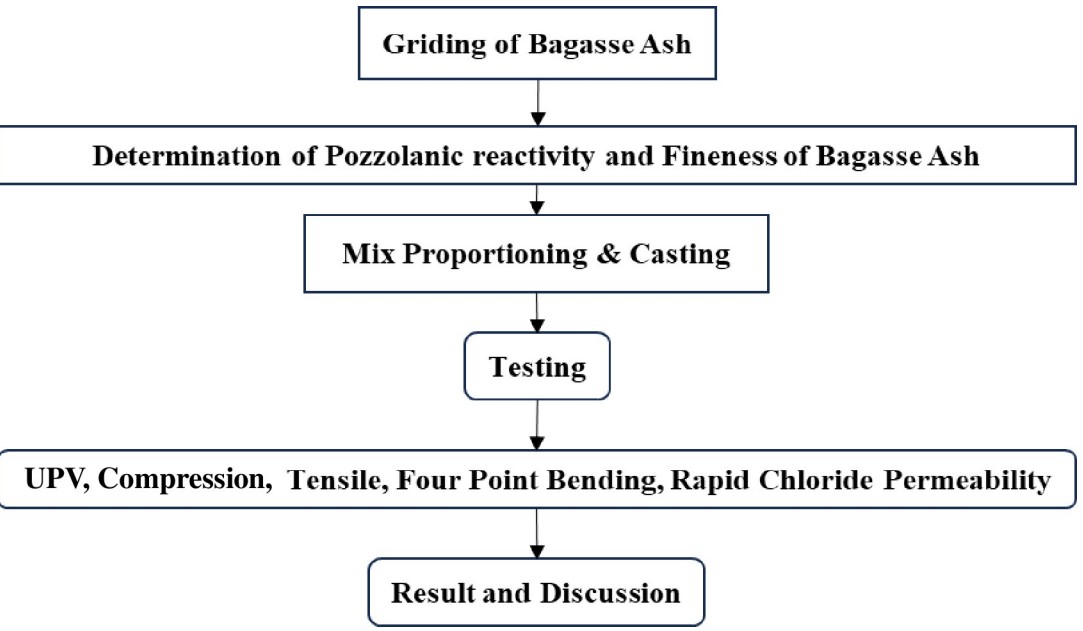

**Fig 3. Schematic of present study.**

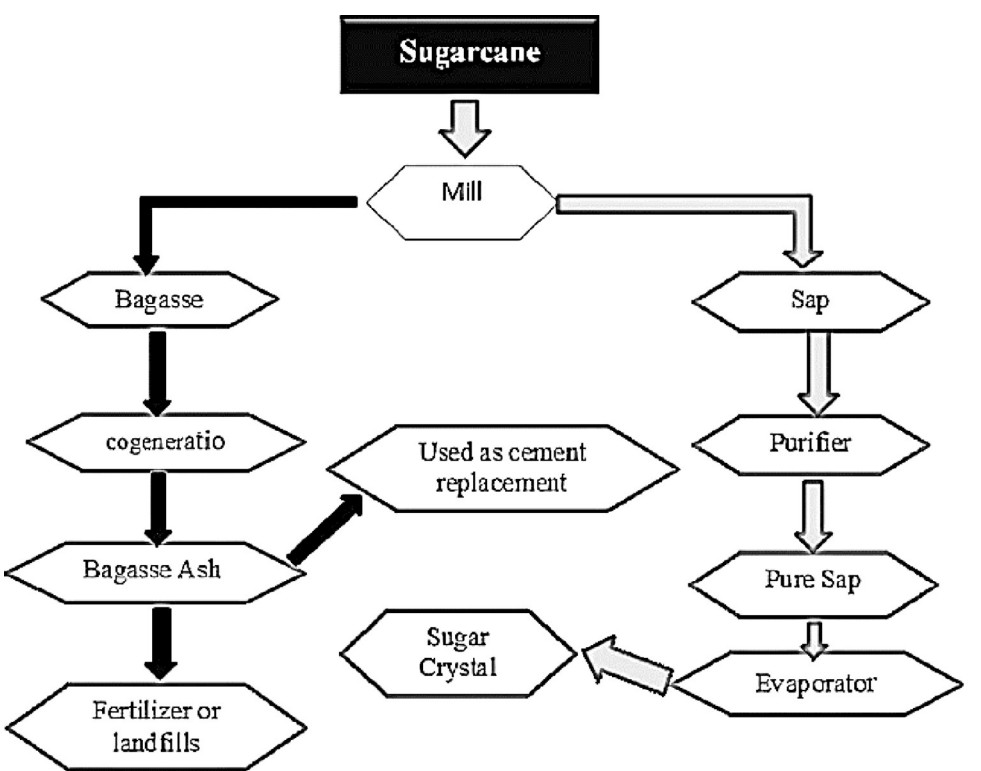

**Fig 4. Extraction of SBA from sugarcane.** Reprinted from [64] under a CC BY license, with permission from construction and building materials, original copyright [2020].

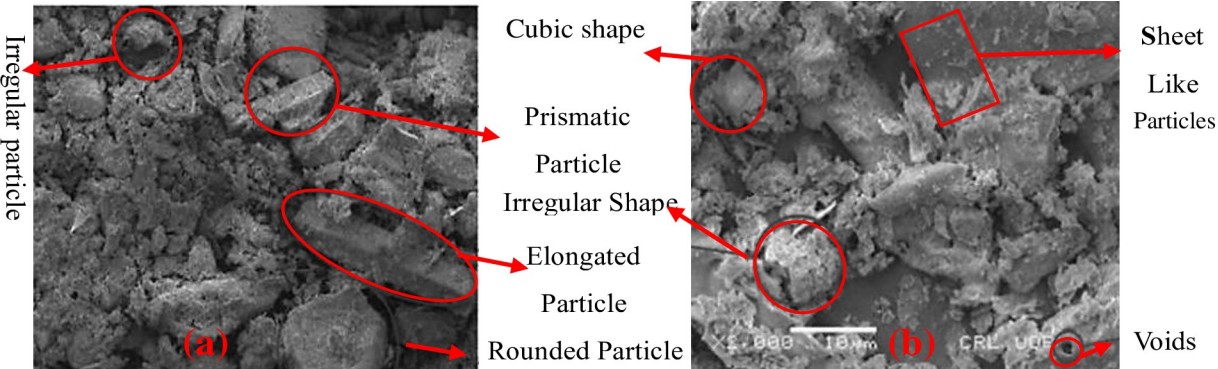

**Fig 5.** GBA after being pulverized in a ball mill: (a) a variety of particle shapes (b) presence of small, uneven, flat particles. Reprinted from [76] under a CC BY license, with permission from frontiers in materials., original copyright [2020].

The micrographs clearly depict a variety of particles in GBA, including round, elongated, irregular, and prismatic forms. The observed particle sizes ranged from 10 to 50 μm. Additionally, the images reveal the existence of small, flat, and pointed particles [76].

## Mix proportions

The production of the AAC mix involves utilizing various components, such as binders, activators, GBA, PVA fibers, and sand. The binder, comprising 20% GGBS and 80% fly ash, was a pivotal element in this investigation. For the formulation of five AAC mixes, varying amounts of class F fly ash (0%, 5%, 10%, 15%, and 20%) were replaced with GBA. Additional water for all AAC mixes was added according to the required flow diameter (120 mm-170 mm), and PVA fibers were incorporated in amounts corresponding to 1.5% of the total mix volume. The alkaline activator was prepared by mixing sodium silicate ($Na_2SiO_3$) solution and sodium hydroxide (NaOH) solution. The detailed mixed design is presented in Table 4. Each AAC mix was assigned a unique label: the control mix devoid of GBA was named C-AAC, while the mixes containing 5%, 10%, 15%, and 20% GBA were designated AAC-5, AAC-10, AAC-15, and AAC-20, respectively.

The selection of the GBA proportions (ranging from 5 to 20%) was based on the findings of previous studies [28, 81]. Workability significantly diminished when GBA levels exceeded 10%. In the absence of GBA, the composite manifested favorable characteristics, spreading readily, and exhibited satisfactory flow. However, upon introducing GBA into the paste, the viscosity increased, especially at 20% GBA content, rendering it so thick that minimal flow was observed. This outcome can be ascribed to the high porosity and rapid absorption rate of the GBA. For casting and drying processes, an ideal workability range of 150–250% is

**Table 4. AAC mix proportions.**

| Sample ID | GGBFS (Kg/m³) | Fly ash (Kg/m³) | GBA (Kg/m³) | Total Water (Kg/m³) | Sand (Kg/m³) | PVA Fibers (Kg/m³) | NaOH (Kg/m³) | Na₂SiO₃ (Kg/m³) | SP (Kg/m³) |
|---|---|---|---|---|---|---|---|---|---|
| **C-AAC** | 130 | 520 | 0 | 215 | 1120 | 25 | 140 | 310 | 0 |
| **AAC-5** | 130 | 487.5 | 32.5 | 230 | 1120 | 25 | 140 | 310 | 2.5 |
| **AAC-10** | 130 | 455 | 65 | 245 | 1120 | 25 | 140 | 310 | 5 |
| **AAC-15** | 130 | 422 | 98 | 255 | 1120 | 25 | 140 | 310 | 7 |
| **AAC-20** | 130 | 390 | 130 | 265 | 1120 | 25 | 140 | 310 | 7.5 |

recommended [82, 83]. Consequently, the optimal concentration of added GBA in the composite has to be below 30%.

## Mixing and curing

To prepare the AAC mix, sand, binders, and GBA were added to the mixer. After the desired quantities were added, the ingredients were thoroughly mixed for a duration of two minutes. Subsequently, activators, water, fibers, and SP were added to the dry mixture and mixed for an additional two minutes to ensure a uniform distribution of PVA fibers throughout the mix. This procedure was based on the methodology developed by Yousefi et al. [84]. The mixtures were then poured into molds and placed in an oven. After curing at 60˚C for 7 days, all specimens were removed from the molds and kept humid at a temperature of 24 ± 2˚C until the day of testing.

## Ultrasonic Pulse Velocity (UPV)

The UPV test was conducted at 7, 14, and 28 days, following the guidelines outlined in ASTM C597–09 [85]. Using two transducers and a pulse-receiver unit with built-in data acquisition, pulse arrival time across samples was measured. Petroleum jelly facilitated transducer-sample coupling. UPV values were calculated by dividing path length by pulse arrival time.

## Mechanical strength test

For each mix, namely C-AAC, AAC-5, AAC-10, AAC-15, and AAC-20, 50 mm cube specimens were cast to study the compressive strengths. The tests were conducted on day 7, 14, and 28. Three identical specimens were tested for each age group. The testing technique followed the guidelines specified in ASTM C109 / C109M-16a standards [86]. The compression machine shown in Fig 6(A) was used to conduct the tests. In accordance with the guidelines outlined in ASTM C109, the test was conducted under a load-controlled mode, and the loading rate was set as 0.25 MPa/s.

To conduct uniaxial tensile tests, nine coupon specimens with dimensions of 152 mm × 76 mm × 13 mm were fabricated for each AAC mix. Experiments were conducted at intervals of

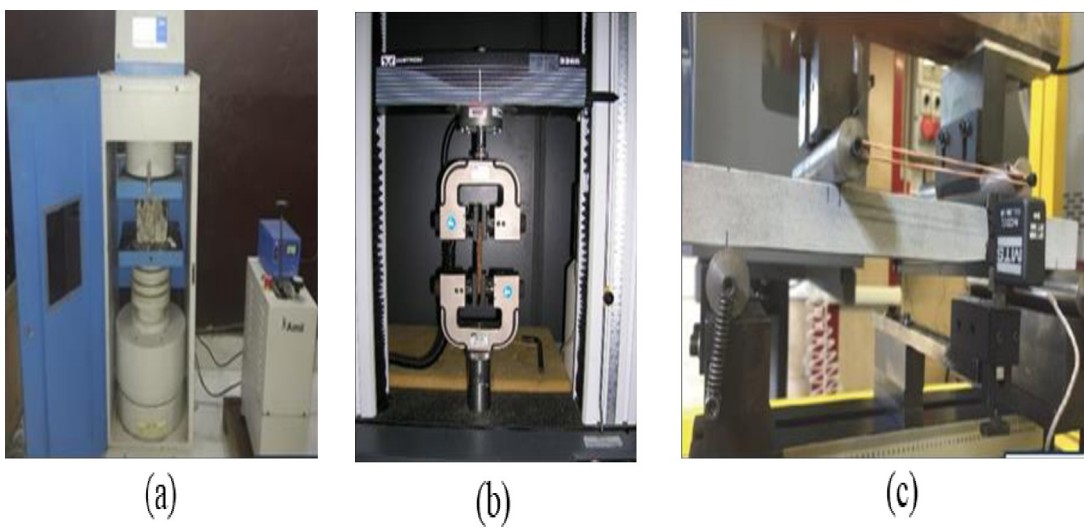

**Fig 6.** (a) Compression testing machine (b) uniaxial tensile testing machine, and (c) flexural testing machine.

7, 14, and 28 days. Direct tensile tests were conducted using a uniaxial tensile testing machine, as shown in Fig 6(B).

The flexural strengths of the AAC mixtures were assessed using a four-point loading method in compliance with ASTM standards [87]. Nine rectangular beams with dimensions of $320 \times 40 \times 12$ (mm) were tested at three different ages: 7, 14, and 28 days.

### Rapid chloride permeability test (RCPT)

In order to assess the durability of concrete and its capacity to resist the penetration of chloride ions, RCPT was performed [88]. RCPT was initially introduced by Whiting in 1981 and has since been standardized by ASTM and AASHTO [89, 90]. In this test, a cylindrical specimen of the saturated composite was subjected to a voltage of 60 Volts. During the experiment, one side of the composite sample was soaked in a 3% NaCl solution and the other side was exposed to a 0.3 M NaOH solution. The resulting current was measured every 30 minutes for a period of six hours. Applying the trapezoidal rule (Eq 1), the total charge passed in the coulombs was calculated. The "charge passed" value was used to classify the composite into distinct categories.

$$Q = 900(I_o + 2I_{30} + 2I_{60} + \ldots + 2I_{330} + I_{360}) \tag{1}$$

In Eq 1 the charge passed is Q (in coulombs), $I_o$ represents the current (in amperes) immediately after applying the voltage, and $I_t$ denotes the current (in amperes) at 't' minutes after applying the voltage. The categories of chloride ion penetrability in concrete based on the charge passed, as specified by the ASTM C1202 standards, are shown in Table 5 [89].

## Results and discussion

### Flowability of AAC

All the composite mixtures flowed easily, exceeding 140 mm diameter when tested according to ASTM C1437. This indicates good cohesion and workability, making them suitable for construction applications. While the spherical nature of fly ash contributes to the "ball-bearing effect" and improves the workability of AAC [91, 92], these mortars generally face inherent workability challenges compared to OPC. This stems from the presence of silicates in their precursors, which are sticky in nature. The inclusion of GBA further exacerbates this issue, as the study found a decrease in flowability with increasing GBA content (Fig 7). As expected, the control mix without GBA exhibited the largest flow diameter. Additionally, all the mixes exhibited a gradual decline in flowability over time, as observed at 0, 5, 10, and 15 min. When the amount of GBA was increased to 20%, the SP dosage also needed to be increased. This is a result of several factors. First, the prismatic shape of the GBA particles increases the friction between the particles and allows a large percentage of voids to be filled with water prior to the flow of the mixture. Second, the GBA particles are porous and absorb some of the mixed water [93].

### Ultrasonic Pulse Velocity

The UPV values for all formulations increased with curing age, indicating progressive development of the internal structure over time. The presented results, depicted in Fig 8, reveal a

**Table 5. Chloride ion penetrability based on charged passed.**

| Chloride ion penetrability | High | Moderate | Low | Very Low | Negligible |
|---|---|---|---|---|---|
| Charged passed (Coulombs) | >400 | 2000–4000 | 1000–2000 | 100–1000 | <100 |

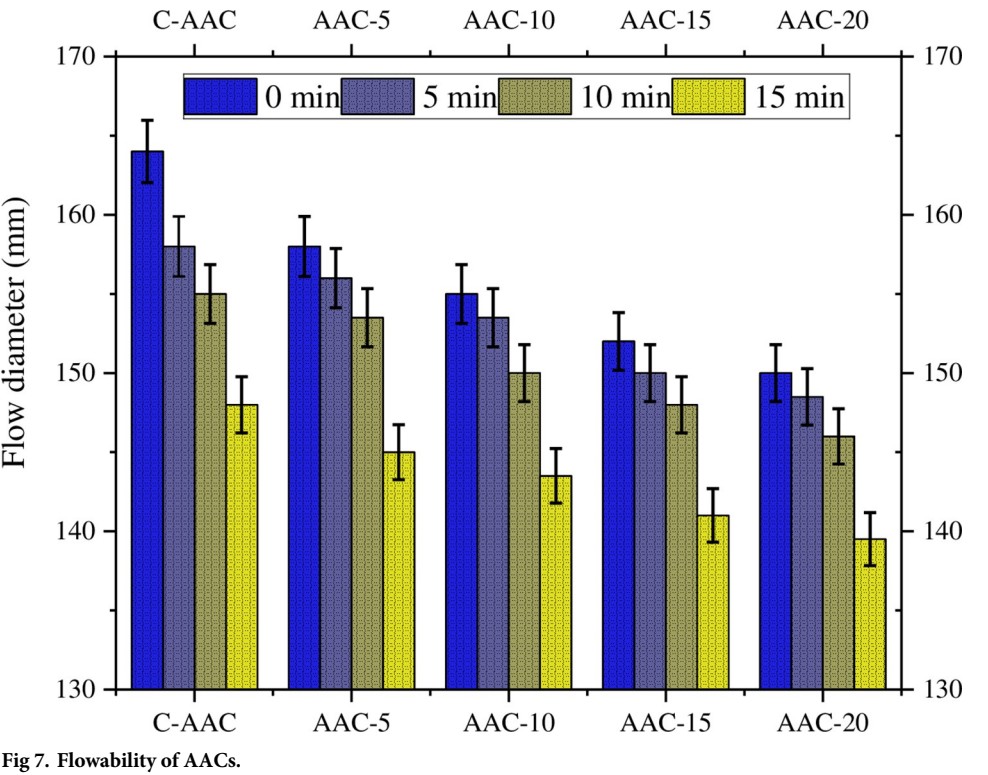

**Fig 7. Flowability of AACs.**

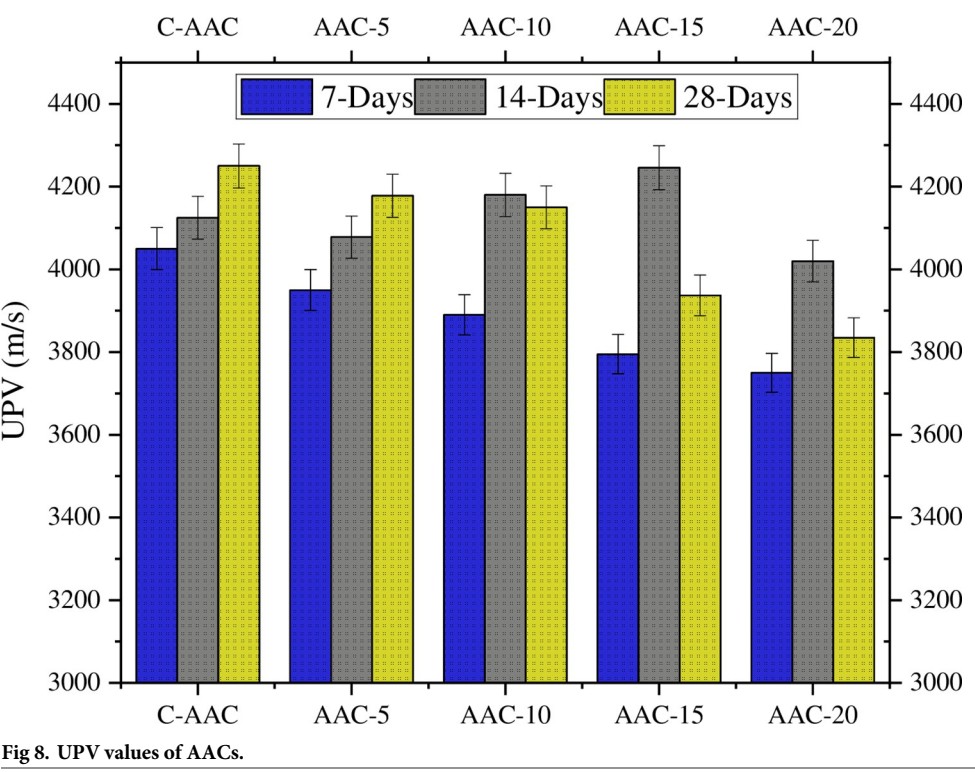

**Fig 8. UPV values of AACs.**

minor adverse effect on the UPV measurements when the GBA content is increased from 0 to 10%. However, a more pronounced effect was observed when the GBA content increased from 10 to 20%. These findings align with the research conducted by Hernández [94], who reported a reduction in the UPV readings with the addition of GBA to mortar mixtures. Additionally, the results indicate that during the initial 14 days, the rate of change in the UPV values was higher than that on the 28th day of testing. Similar findings have been reported in other investigations [95, 96]. Akbar et al. studied the effect of fiber addition on the UPV values and noted that the velocity of ultrasonic waves was higher through AACs without any fibers. The inclusion of polypropylene (PP) fibers increased the porosity of the composite, leading to a decrease in the UPV as the PP fiber content increased [63].

## Compressive strength

The influence of GBA on the fresh and hardened properties of the alkali-activated matrix is intricately related to its size and shape. Fig 10(A) presents a comparative analysis of the compressive strength–obtained from the average of three samples–across the four AAC mixes. These mixes featured varied proportions of GBA and were assessed at curing ages of 7, 14, and 28 days. During the early stages of curing (7 and 14 days), all AAC mixes incorporating GBA demonstrated lower strength than C-AAC because of the limited or insignificant pozzolanic reactivity of GBA during the initial phases. The reduction in strength was more pronounced with higher GBA content. The decrease in strength with increasing GBA content can also be associated with the presence of highly crystalline quartz ($SiO_2$) in the GBA. The stable quartz content in GBA remains largely undissolved in the alkaline solution, hindering the completion of the deconstruction step (explained below) and impeding polymerization. The larger particle size of GBA, exceeding that of raw fly ash by more than 10 times, further contributes to its reduced reactivity [97].

The lower strengths of all AACs at 7 and 14 days can be explained in terms of the alkali activation process comprising three pivotal stages: deconstruction, polymerization, and stabilization [98]. Initially, alumina and silica dissolved from fly ash in the alkaline-activated solution, leading to polymerization and the formation of aluminosilicate polymers. The subsequent stabilization phase involves the interconnection of these gel structures, contributing to the development of paste strength. However, this stabilization process is inherently slow, necessitating an extended duration for strength to reach its full potential, thereby elucidating the lower strengths of all AACs at early ages [99].

After 28 days, the decline in strength associated with increasing GBA content stabilized, with the highest strength value achieved when the GBA content was 10%. Similar behavior, characterized by diminishing strength during early ages and escalating strength during later stages, have been reported for engineered cementitious composite (ECC) containing GBA [76, 100, 101].

The decreased strength of ACC-20 compared to ACC-10 can be attributed to the low content of fly ash and higher content of GBA, which contains a significant amount of crystalline quartz ($SiO_2$). The initial sources of alumina and silica must first dissolve in an alkali activated solution before subsequent alkali activation stages can proceed. Therefore, it is preferable for these sources to be in an amorphous form to facilitate dissolution. This is the case with raw fly ash during its activation process. However, due to the presence of quartz in GBA, which is highly stable and resistant to dissolution in alkaline solutions, the breakdown step is difficult to achieve, resulting in limited activation. Consequently, GBA does not contribute much to the formation of aluminosilicate gels. Another factor is the disparity in particle size between raw fly ash and GBA, with GBA particles being larger, rendering them less reactive and thus negatively impacting strength when used in higher quantities [102].

In a study conducted by Akbar et al., it was observed that at 28 days, bagasse ash based AAC without any fiber inclusion exhibited a strength 3.6% higher than that of a control cement formulation. He introduced 1% PP fibers and observed a maximum improvement of 5.8%, which was attributed to the crack arrest capability of the PP fibers. Moreover, an increase in the PP fiber content (2% and 3%) led to a reduction in compressive strength, although the strength remained higher than that of the cement counterpart [63].

In another investigation carried out on OPC-based mortar, Arenas-Piedrahita et al. reported that the compressive strength of mortars containing 10% untreated fly ash resembled that of mortars with 10% untreated GBA. However, as the proportion of untreated fly ash increased from 10 to 20%, the compressive strength decreased [101]. Other studies have suggested that exceeding 10% substitution of cement by GBA can adversely impact the compressive strength of OPC mortars and concretes [28, 103].

## Tensile strength

The uniaxial tensile strengths of the AAC mixes at curing ages of 7, 14, and 28 days are shown in Fig 9(B). After 7 and 14 days of curing, the AAC mixes with GBA exhibited reduced tensile strength compared to C-AAC. At 28 days, the tensile strength increased with increasing GBA content up to 10%, beyond which the strength continued to decrease. A similar increasing trend followed by decreasing values was observed by Sousa et al. [104]. The reduction in strength can be attributed to the higher amount of stable quartz with increasing GBA content beyond 10% [42]. At 28 days of curing, AAC-10 demonstrated 8.6% higher tensile strength than C-AAC and was recommended for producing optimized AACs.

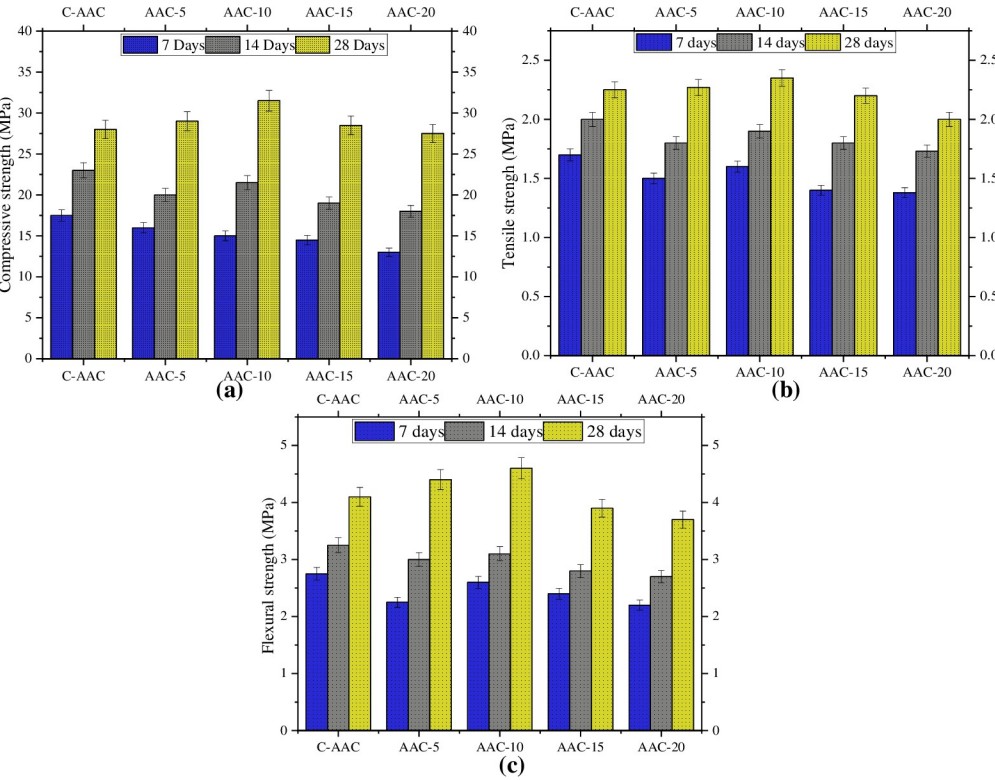

**Fig 9.** Comparison of strength at 7, 14, and 28 days (a) Compressive strength (b) Tensile strength (c) Flexural strength.

Similar to PVA fibers, PP fibers as studied by Akbar et al., have proven to be an effective means to augment the tensile strength of bagasse ash based AAC. The tensile strength of AAC without PP fibers exhibited a negligible increase (0.78% higher) compared to that of the cement mortar. However, when fibers were added, 40 to 50% improvement was noticed. This improvement can also be attributed to the denser microstructure of the bagasse ash-based composite, which facilitates effective stress transfer to the fibers and leads not only to improved tensile strength but also to enhanced capabilities in crack arrest and propagation within the composite [63, 105].

## Flexural strength

In Fig 9(C), the flexural strength of AAC is depicted at time intervals of 7, 14, and 28 days. For GBA contents, flexural strength increases in later stages, like the phenomena observed in tension and compression. The gradual decline in strength beyond 10% GBA content can be associated with a higher amount of stable quartz and lower amount of fly ash as previously discussed. Nasir et al. displayed analogous behavior for ECC containing GBA [76].

In an investigation by Akbar et al., bagasse ash based AAC demonstrated a noteworthy enhancement in flexural strength. Their study revealed that the influence of PP fiber inclusion in AAC was particularly pronounced with a 2% addition of PP fibers, with approximately 17% and 42% improvements compared to plain AAC and plain cement mortar, respectively. However, further escalation of the PP fiber content adversely affected the flexural strength. The reduction in flexural strength is attributed to the poor dispersion of the fibers at higher dosages [63].

## Strength development

Fig 10 depicts the strength activity index (SAI) values, representing the change in the composite strength compared with the control mix. According to ASTM C618 [106], it is recommended that the SAI value should exceed 75% for any pozzolanic material used as a cement substitute. This criterion is met by all mixes at all ages, except for the 7th-day compressive strength of AAC-20. Across all ages and for all types of strength (compressive, tensile, and flexural), the SAI values exhibit an increasing trend up to AAC-10, mirroring the pattern observed in the 28-day strength, beyond which there is a decline. Notably, AAC-10 demonstrated the highest SAI value at 28 days, demonstrating its superior strength performance.

Fig 11 provides a comprehensive overview of the development of compressive, tensile, and flexural strengths over time. The values marked on the lines corresponding to the 7th, 14th, and 28th days indicate the percentage increase or decrease in strength relative to C-AAC of the same age. The slope of the line between any two mixes indicates whether the percentage variation in the strength increases or decreases as we move from one sample to another. In Fig 11 (A), it is evident that at 28 days, AAC-5, AAC-10, and AAC-15 displayed compressive strengths 3.16%, 10%, and 1% higher, respectively, than C-AAC. Conversely, AAC-20 exhibited lower compressive strength across all ages. The 28th-day strength shows higher values for GBA content up to 15% in tension, compression, and flexure. However, on the 7[th] and 14[th] days, the strengths for all GBA percentages were lower than those for C-AAC. The steep negative slope of the 7th-day line signifies a pronounced decline in strength at initial ages.

## Rapid chloride permeability test

In the RCPT, the C-AAC exhibited a total charge of 2800 coulombs at 28 days and 2725 coulombs at 56 days, indicating 'moderate' resistance to chloride ion ingress as per ASTM C1202-12 [89]. Addition of GBA resulted in a substantial reduction in the electrical conductance, as

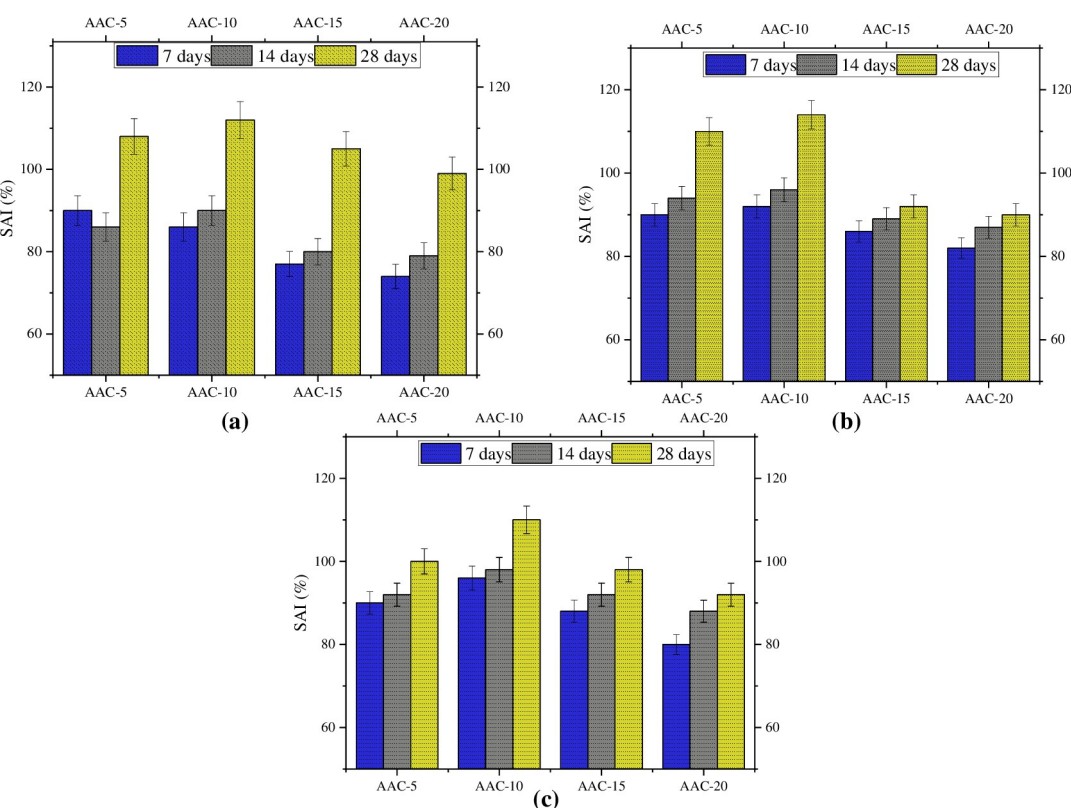

**Fig 10.** SAI values of mixes at 7 days, 14 days, and 28 days (a) Compressive strength (b) Tensile strength (c) Flexural strength.

illustrated in Fig 12. Compared to C-AAC, specimens with 10% and 15% addition showed a 20% and 32% decrease in the total charge at 56 days, respectively. All GBA-incorporated composite exhibited higher resistance than C-AAC at 28 and 56 days, categorizing them as 'low' permeable composites as per ASTM guidelines. Supplementary materials, as emphasized by Shi et al., diminishes pore connectivity and reduces concrete permeability [107]. The reduction in current flow can also be attributed to other factors, including the use of cementing materials containing silica, reduced pore solution conductivity (due to a decrease in $Na^+$, $K^+$, $Ca^{++}$, and $OH^-$ ion concentrations), and an improved pore structure resulting from GBA's pozzolanic performance. After 28 and 90 days of curing, Ganesan et al. [28] demonstrated a 50% decrease in current flow, relative to the control, in concrete incorporating GBA. Arenas-Piedrahita et al. [101], conducted a study where OPC-based mortar, integrating GBA and fly ash, showcased a decrease in the overall current passed over time due to ongoing hydration and pozzolanic reactions. The findings highlighted a favorable influence of GBA inclusion on mortar impermeability across all ages.

## Economic and environmental assessment

In this study, GBA, GGBS, and fly ash were used instead of OPC, resulting in cost savings in the concrete mix. The cost per kilogram in Pakistani rupees (PKR) for various raw materials is presented in Table 6. Cost reduction can have significant benefits for large-scale construction

**Table 6. Unit cost of raw materials for concrete.**

| Raw Materials | Cement | GGBS | GBA | Fly Ash | Na₂SiO₃ | NaOH | Fine Aggregate | PVA Fibers |
|---|---|---|---|---|---|---|---|---|
| Cost (PKR/Kg) | 30 | 10 | 14 | 7 | 130 | 400 | 1 | 1050 |

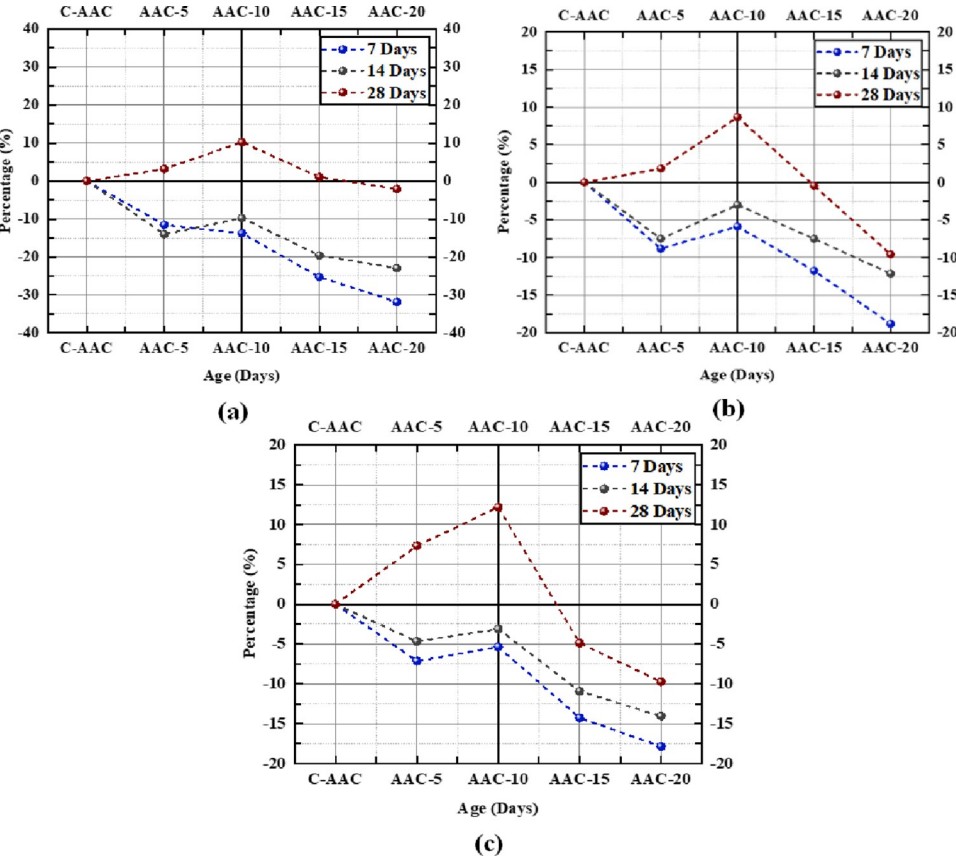

**Fig 11.** Percentage variation in strength for varying GBA content (a) Compressive strength (b) Tensile strength (c) Flexural strength.

projects, such as dams and multistory buildings, where concrete production costs constitute a substantial portion of the total expenses [108]. The addition of fibers may be costly, but this increment in cost is deemed acceptable when prioritizing the improvement of mechanical properties.

The use of waste materials is an effective way to reduce natural resource usage and greenhouse gas emissions. $CO_2$ emissions from each composite mixture can be estimated using Eq 2 [109, 110]. Emission factors were determined from the literature for each component and are listed in Table 7. As can be seen, cement has much higher values of $CO_2$ emission factor as

**Table 7. $CO_2$ emissions for different concrete component.**

| Concrete component | EF (kg-$CO_2$/ton) |
|---|---|
| **OPC** [110] | 650 |
| **GBA** [111] | 25 |
| **GGBS** [112] | 182 |
| **Fly ash** [113] | 0.023 |
| **NaOH** [114] | 1915 |
| **Na2SiO3** [114] | 1514 |
| **Fine aggregate** [110] | 4.6 |
| **Concrete mixer** [110] | 1.61 |

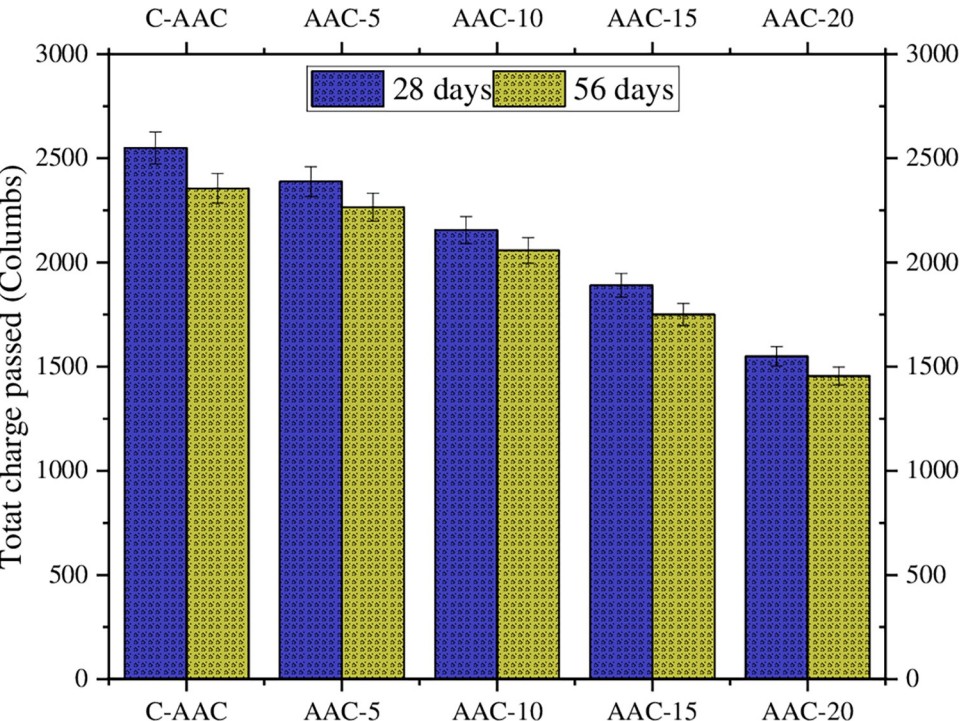

**Fig 12. Total charge passed at 28 and 56 days of curing.**

compared to other binders whose emissions are primarily due to transportation and processing. Therefore, the use of GBA and fly ash in concrete offers technical, economic, and sustainability benefits to the construction sector.

$$EF_{concrete} = \sum w \times EF + EF_{mixer} \tag{2}$$

To understand the overall sustainability impact of the composite, it is crucial to assess the leaching behavior of composites containing GBA [115]. Andreão et al. found that the GBA composite exhibited lower leaching than municipal and hospital waste ash [116]. Cement replacement with GBA was found to improve the pore size, making it less susceptible to environmental hazards related to heavy metal leaching. Several other studies have demonstrated that the GBA composite exhibited greater resistance to sulfate ion attacks, making it an ideal material for cement replacement [117].

## Conclusion

From results and discussion sections, the following conclusions can be made.

- The compressive strength of the alkali-activated composite (AAC) at 7 days decreased with increasing ground bagasse ash (GBA) content; however, at later ages, up to 10% GBA content, the compressive strength increased. At 28 days, AAC-10 exhibited a 10% higher compressive strength than C-AAC of the same age.

- The tensile and flexural strengths exhibited a trend like that of the compressive strength. At 28 days, only AAC-5 and AAC-10 demonstrated higher tensile and flexural strengths (8.6% and 12.2% higher, respectively) than C-AAC of the same age. AAC-20 consistently displayed

lower strength than C-AAC across all ages. Notably, on days 7th and 14th days, the strengths for all GBA percentages were lower than those of C-AAC.

- All samples, except AAC-20, showed higher strength activity index (SAI) values for compression, tension, and flexure. Thus, to produce sustainable AAC, any percentage of cement in the range of 0 to 20 can be safely replaced with GBA. Moreover, 10% GBA can be considered the best replacement percentage.

- The incorporation of GBA significantly reduces the electrical conductance of the composite, thereby enhancing its durability and resistance to chloride penetration.

- All AAC formulations are cost-effective and produce less $CO_2$ compared to ordinary Portland cement. The addition of fibers may be costly, but this increment in cost is deemed acceptable when prioritizing the improvement of mechanical properties.

## Recommendations

Here are the recommendations for future research.

- Other durability tests on AAC containing GBA and polyvinyl alcohol (PVA) fibers should be conducted. This can include chloride diffusion and acid attack tests.

- The performance of AAC containing GBA should be assessed over extended periods, considering its resistance to freeze–thaw cycles, shrinkage, and creep.

- The influence of varying the PVA fiber percentage on the mechanical properties of AAC should be assessed.

## Supporting information

**S1 File. Data obtained in this study.**
(PDF)

## Acknowledgments

The authors would like to thank the Department of Civil Engineering, Ghulam Ishaq Khan Institute of Engineering Sciences and Technology Pakistan for providing support in this research.

## Author Contributions

**Conceptualization:** Munir Iqbal.

**Funding acquisition:** Loai Alkhattabi, Hisham Alabduljabbar.

**Investigation:** Munir Iqbal, Loai Alkhattabi, Sohaib Nazar, Jihad Alam, Hisham Alabduljabbar, Ali Husnain.

**Methodology:** Munir Iqbal, Loai Alkhattabi, Jihad Alam, Hisham Alabduljabbar.

**Project administration:** Muhammad Ashraf.

**Resources:** Muhammad Ashraf.

**Supervision:** Muhammad Ashraf.

**Validation:** Munir Iqbal, Jihad Alam, Ali Husnain.

**Writing – original draft:** Munir Iqbal.

**Writing – review & editing:** Muhammad Ashraf, Sohaib Nazar.

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
