## [Decision Letter · Decision Letter 0]

5 Jun 2024

PONE-D-24-19129Development of sustainable alkali activated composite incorporated with sugarcane bagasse ash and polyvinyl alcohol fibers.PLOS ONE

Dear Dr. Iqbal,

Thank you for submitting your manuscript to PLOS ONE. After careful consideration, we feel that it has merit but does not fully meet PLOS ONE’s publication criteria as it currently stands. Therefore, we invite you to submit a revised version of the manuscript that addresses the points raised during the review process.

We look forward to receiving your revised manuscript.

Kind regards,

André Gustavo de Sousa Galdino

Academic Editor

PLOS ONE

Journal Requirements:

2. We note that your Data Availability Statement is currently as follows: 

"All relevant data are within the manuscript and its Supporting Information files."

Reviewers' comments:

Reviewer's Responses to Questions

**Comments to the Author**

1. Is the manuscript technically sound, and do the data support the conclusions?

Reviewer #1: Yes

Reviewer #2: Yes

2. Has the statistical analysis been performed appropriately and rigorously? 

Reviewer #1: N/A

Reviewer #2: N/A

3. Have the authors made all data underlying the findings in their manuscript fully available?

Reviewer #1: Yes

Reviewer #2: Yes

4. Is the manuscript presented in an intelligible fashion and written in standard English?

Reviewer #1: Yes

Reviewer #2: Yes

5. Review Comments to the Author

Reviewer #1: Manuscript entitled as "Development of sustainable alkali activated composites incorporated with sugarcane bagasse ash and polyvinyl alcohol fibers" discuss about the usage of sugar cane bagasse ash in geopolymer concrete. To improve the quality of manuscript authors are requested to address the following comments

1. Line 16: it is "mechanical and durability properties"

2. Rewrite the research gap

3. Include specific gravity of all materials (fly ash, GBA, etc)

4. Include particle size distribution curve for GBA and fly ash

5. Include characteristics of superplastizer

6. Check line 146. Include sample preparation for SEM

7. Table 4: is it additional water or water? Check it

8. Include information about molarity in the alkalis used in the mix

9. Fig 7 & 8, variations in the graphs are not visible. So check the axis scale

10. Strength activity index and Compressive Strength should not be in separate sub heading. Include any one part

11. Include conclusion about section 4.9 in conclusion

12. To improve the quality of manuscript authors may consider following references

a. https://doi.org/10.1016/j.jobe.2022.105235

b. https://doi.org/10.1007/s12046-015-0390-6

c. https://doi.org/10.1007/s12046-022-01963-7

d. https://doi.org/10.1007/s40030-019-00359-x

e. https://doi.org/10.1016/j.cscm.2023.e02435

f. https://doi.org/10.1016/j.jobe.2023.107836

Reviewer #2: This study was aimed to study the Development of sustainable alkali activated composite incorporated with sugarcane bagasse ash and polyvinyl alcohol fibers but there are some issues needed to be clarified before it can be accepted for publication

1-This is an interesting study, and the authors present a very substantial study, which the reviewers support for further study.

2- All results should be displayed as figures

3-error bars must be added to the figures.

4-Figures must have high resolution

5-More discussion in depth must be made with the related literature

-10.1016/j.scp.2023.101319

-10.1016/j.scp.2024.101512

-10.1016/j.jobe.2023.106661

-10.1016/j.conbuildmat.2023.134655

6. PLOS authors have the option to publish the peer review history of their article (what does this mean?). If published, this will include your full peer review and any attached files.

Reviewer #1: No

Reviewer #2: No

---

## [Author Response · Author response to Decision Letter 0]

12 Jun 2024

PONE-D-24-19129

DEVELOPMENT OF SUSTAINABLE ALKALI ACTIVATED COMPOSITE INCORPORATED WITH SUGARCANE BAGASSE ASH AND POLYVINYL ALCOHOL FIBERS.

JOURNAL REQUIREMENTS

1. Please ensure that your manuscript meets PLOS ONE's style requirements, including those for file naming. The PLOS ONE style templates can be found at ……………………..

Response: Thank you very much for taking the time to review our manuscript and for providing valuable feedback. We truly appreciate the effort and expertise that went into evaluating our work. To meet PLO ONE's style requirement, we have carefully reviewed the provided templates and have made the necessary changes to ensure compliance. Regarding file naming, the marked-up copy of our manuscript, highlighting changes made to the original version, has been uploaded as a separate file labeled 'Revised Manuscript with Track Changes'. As per suggestions, we have also included an unmarked version of our revised paper labeled as 'Manuscript'.

2. We note that your Data Availability Statement is currently as follows: 

"All relevant data are within the manuscript and its Supporting Information files."

Please confirm at this time whether or not your submission contains all raw data required to replicate the results of your study……………………….

Response: Thank you for giving us the opportunity to clarify this.

We confirm that all relevant data are within the manuscript and its Supporting Information files. Our study involved conducting lab experiments, and the data gathered from these experiments are thoroughly presented in the manuscript in both tabular and graphical form. 

COMMENTS TO THE AUTHOR

1. Is the manuscript technically sound, and does the data support the conclusions?

Reviewer #1: Yes

Reviewer #2: Yes

2. Has the statistical analysis been performed appropriately and rigorously?

Reviewer #1: N/A

Reviewer #2: N/A

3. Have the authors made all data underlying the findings in their manuscript fully available?

Reviewer #1: Yes

Reviewer #2: Yes

4. Is the manuscript presented in an intelligible fashion and written in standard English?

Reviewer #1: Yes

Reviewer #2: Yes

REVIEW COMMENTS TO THE AUTHOR

REVIEWER #1: 

Manuscript entitled "Development of sustainable alkali activated composites incorporated with sugarcane bagasse ash and polyvinyl alcohol fibers" discuss about the usage of sugar cane bagasse ash in geopolymer concrete. To improve the quality of manuscript authors are requested to address the following comments:

1. Line 16: it is "mechanical and durability properties"

Response: Thank you for your valuable feedback. We have corrected the error as suggested. The sentence now reads: "This study aims to develop and enhance the mechanical and durability properties of alkali-activated composites (AACs) by incorporating varying amounts..."

Thank you again for your careful review.

2. Rewrite the research gap

Response: Thank you for your insightful feedback. We have enhanced our discussion on the latest research developments related to our research area. This comprehensive discussion is included in the second-to-last paragraph of the introduction section. Additionally, in the final paragraph of the introduction, we have provided a concise description of the research gap and explained how our study addresses it. Here is a glimpse of it:

The research gap we identified is the unexplored behavior of untreated ground bagasse ash (GBA) in AACs containing fly ash, GGBS, and PVA fibers. Our study addresses this gap by evaluating AACs with varying proportions of untreated GBA which allowed us to identify the optimal amount of GBA for improved mechanical performance and durability.

All changes have been highlighted in yellow for your review. Thank you again for your valuable suggestions.

3. Include specific gravity of all materials (fly ash, GBA, etc)

Response: Thank you for identifying this critical point. We have now included a separate column for the specific gravity of each material (fly ash, GBA, etc.) in Table 2. The changes have been highlighted in yellow for easy identification.

4. Include particle size distribution curve for GBA and fly ash

Response: Thank you for highlighting this important point. Based on your suggestion, we have now included the particle size distribution curve of the GBA in Figure 2.

5. Include characteristics of superplastizer

Thank you for raising this important point. As per your suggestion, we have included a discussion on the properties of plasticizers. This addition can be found under the subheading 'Activator and Superplasticizer.' The changes are reproduced below for your convenience:

“To induce flowability and reduce slump retention, a sulfonated naphthalene-based water-reducing agent conforming to ASTM C - 494 Type G was used as a superplasticizer (SP). It was a liquid with a dark brown color and had a density of 1.13 ± 0.03 g/cm³. The recommended dosage ranged from 0.3% to 2%. This SP contained no chlorides, ensuring it was suitable for use in reinforced concrete without the risk of corrosion. It had a pH value of 4.8 and a solids content of 30%.”

6. Check line 146. Include sample preparation for SEM

Thank you for mentioning this key point. The SEM images included in this manuscript are taken from another research work, and the reference to which is given in the relevant paragraphs. We have now added a separate sentence in the captions stating that the images are reproduced under the CC license from another publisher, and the reference to the original article and publisher is also provided in the caption. Since these SEM images are from another study, it would not be relevant to include the sample preparation details in our manuscript. 

7. Table 4: is it additional water or water? Check it

We appreciate your careful review and valuable feedback.

We would like to clarify that the term "additional water" used in our manuscript was mistakenly used for the total water content, which includes both the water in the alkali activators solution and the extra water added to achieve the desired workability. We have addressed this mistake, and in our revised manuscript, it is now "total water" instead of "additional water."

8. Include information about molarity in the alkalis used in the mix.

Thank you for highlighting the importance of the molarity of the activator solution. The molarity of the NaOH solution is 14 moles, and the concentration of sodium silicate is given in weight percent, as detailed in the manuscript under the section titled 'Activators and Superplasticizers.' For your convenience, the relevant text is reproduced below:

“In this study, the alkaline activator was prepared by mixing sodium silicate (Na2SiO3) solution and sodium hydroxide (NaOH) solution. The Na2SiO3 solution comprised of 14.7% sodium oxide (Na2O), 29.4% silicate (SiO2), and 44.1% solids. The NaOH solution with a molarity of 14 was obtained by dissolving solid granulated caustic soda in tap water.”

9. Fig 7 & 8, variations in the graphs are not visible. So, check the axis scale

We appreciate your feedback.

Regarding your suggestions on Figures 7 and 8, we have scaled down the Y-axis to enhance clarity and depict variations among different mixes more effectively. Figure 7 now starts from 130 mm (previously 0 mm), and Figure 8 starts from 3000 m/s (previously 0 m/s). These adjustments make the variations more pronounced, and we have also added error bars to each graph for improved data representation.

10. Strength activity index (SAI) and Compressive Strength should not be in separate sub heading. Include any one part.

Thank you for helping us improve the organization and coherency of our manuscript. We've made the necessary revisions to the manuscript. Now, the discussion on SAI comes after the section covering compression, tensile, and flexural strength.

We've also merged the subheadings "SAI" and "percentage variation in strength" into one subheading called "strength development." This change makes more logical sense as it follows the discussion on the strength of the AACs and delves into how the strength evolves over time. Additionally, through SAI values, we can check whether or not the ACCs meet the threshold set by ASTM C618.

--- 11. Include conclusion about section 4.9 in conclusion

Thank you for bringing up this important point. As per your suggestion, we have included a conclusion about the environmental and economic assessment in the conclusion section of our manuscript. The change has been highlighted in yellow for easy identification.

12. To improve the quality of manuscript authors may consider following references

a. https://doi.org/10.1016/j.jobe.2022.105235

b. https://doi.org/10.1007/s12046-015-0390-6

c. https://doi.org/10.1007/s12046-022-01963-7

d. https://doi.org/10.1007/s40030-019-00359-x

e. https://doi.org/10.1016/j.cscm.2023.e02435

f. https://doi.org/10.1016/j.jobe.2023.107836

Response: Thank you for guiding us towards these highly relevant articles. We have thoroughly examined each of the mentioned articles and incorporated pertinent information from them into our manuscript. Additionally, we have ensured that all these sources are properly cited. Your insightful recommendation has greatly enriched the depth of our discussion, and we are sincerely grateful for your invaluable assistance.________________________________________

REVIEWER #2:

This study was aimed to study the Development of sustainable alkali activated composite incorporated with sugarcane bagasse ash and polyvinyl alcohol fibers but there are some issues needed to be clarified before it can be accepted for publication

1-This is an interesting study, and the authors present a very substantial study, which the reviewers support for further study.

Response: Thank you very much for taking the time to review our manuscript and for providing valuable feedback. Special Thanks for appreciating our work, means a lot for us.

2- All results should be displayed as figures

Response: We genuinely appreciate your keen insight and attention to detail. In response to your valuable suggestion, we have meticulously reviewed our manuscript and ensured that all pertinent results are now presented in graphical form.

3-error bars must be added to the figures.

Response: We genuinely appreciate your keen insight and attention to detail. We have thoroughly revised all the figures in the manuscript and added error bars to all the presented graphs. Your suggestion has significantly enhanced the precision and clarity of our data representation, and we are sincerely grateful for your insightful recommendation.

4-Figures must have high resolution

Response: We completely agree with your observation. Considering your valuable suggestion, we have redrawn all the figures and significantly improved their pixel quality to ensure high resolution.

5-More discussion in depth must be made with the related literature

-10.1016/j.scp.2023.101319

-10.1016/j.scp.2024.101512

-10.1016/j.jobe.2023.106661

-10.1016/j.conbuildmat.2023.134655

Response: Thank you for guiding us towards these highly relevant articles. We have thoroughly examined each of the mentioned articles and incorporated pertinent information from them into our manuscript. Additionally, we have ensured that all these sources are properly cited. Your insightful recommendation has greatly enriched the depth of our discussion, and we are sincerely grateful for your invaluable assistance.________________________________________

Thanks again for all valuable suggestions. We are open to further suggestions and welcome any additional feedback you may have.

---

## [Decision Letter · Decision Letter 1]

1 Jul 2024

Development of sustainable alkali activated composite incorporated with sugarcane bagasse ash and polyvinyl alcohol fibers.

PONE-D-24-19129R1

Dear Dr. Iqbal,

We’re pleased to inform you that your manuscript has been judged scientifically suitable for publication and will be formally accepted for publication once it meets all outstanding technical requirements.

Kind regards,

André Gustavo de Sousa Galdino

Academic Editor

PLOS ONE

Additional Editor Comments (optional):

Reviewers' comments:

Reviewer's Responses to Questions

**Comments to the Author**

1. If the authors have adequately addressed your comments raised in a previous round of review and you feel that this manuscript is now acceptable for publication, you may indicate that here to bypass the “Comments to the Author” section, enter your conflict of interest statement in the “Confidential to Editor” section, and submit your "Accept" recommendation.

Reviewer #1: All comments have been addressed

Reviewer #2: All comments have been addressed

2. Is the manuscript technically sound, and do the data support the conclusions?

Reviewer #1: Yes

Reviewer #2: Yes

3. Has the statistical analysis been performed appropriately and rigorously? 

Reviewer #1: N/A

Reviewer #2: Yes

4. Have the authors made all data underlying the findings in their manuscript fully available?

Reviewer #1: Yes

Reviewer #2: Yes

5. Is the manuscript presented in an intelligible fashion and written in standard English?

Reviewer #1: Yes

Reviewer #2: Yes

6. Review Comments to the Author

Reviewer #1: Authors had addressed all comments in the manuscript in propert manner. Hence, I accept the manuscript for the publication.

Reviewer #2: The authors have responded to the comments of the reviewers satisfactorily.and it can be accepted in the current form

7. PLOS authors have the option to publish the peer review history of their article (what does this mean?). If published, this will include your full peer review and any attached files.

Reviewer #1: No

Reviewer #2: No

---

## [Editor Report · Acceptance letter]

6 Aug 2024

PONE-D-24-19129R1 

PLOS ONE

Dear Dr. Iqbal, 

I'm pleased to inform you that your manuscript has been deemed suitable for publication in PLOS ONE. Congratulations! Your manuscript is now being handed over to our production team.

Kind regards, 

on behalf of

Dr. André Gustavo de Sousa Galdino 

Academic Editor

PLOS ONE